# Improved *V*_th_ Stability and Gate Reliability of GaN-Based MIS-HEMTs by Employing Alternating O_2_ Plasma Treatment

**DOI:** 10.3390/nano14060523

**Published:** 2024-03-14

**Authors:** Xinling Xie, Qiang Wang, Maolin Pan, Penghao Zhang, Luyu Wang, Yannan Yang, Hai Huang, Xin Hu, Min Xu

**Affiliations:** State Key Laboratory of ASIC and System, School of Microelectronics, Fudan University, Shanghai 200433, China; 21212020013@m.fudan.edu.cn (X.X.); 21112020111@m.fudan.edu.cn (Q.W.); mlpan21@m.fudan.edu.cn (M.P.); phzhang19@fudan.edu.cn (P.Z.); wangly20@fudan.edu.cn (L.W.); yangyn20@fudan.edu.cn (Y.Y.); 22212020008@m.fudan.edu.cn (H.H.); 22212020078@m.fudan.edu.cn (X.H.)

**Keywords:** AlGaN/GaN MIS-HEMT, threshold voltage stability, gate reliability

## Abstract

The *V*_th_ stability and gate reliability of AlGaN/GaN metal–insulator–semiconductor high-electron-mobility transistors (MIS-HEMTs) with alternating O_2_ plasma treatment were systematically investigated in this article. It was found that the conduction band offset at the Al_2_O_3_/AlGaN interface was elevated to 2.4 eV, which contributed to the suppressed gate leakage current. The time-dependent dielectric breakdown (TDDB) test results showed that the ALD-Al_2_O_3_ with the alternating O_2_ plasma treatment had better quality and reliability. The AlGaN/GaN MIS-HEMT with the alternating O_2_ plasma treatment demonstrated remarkable advantages in higher *V*_th_ stability under high-temperature and long-term gate bias stress.

## 1. Introduction

AlGaN/GaN metal–insulator–semiconductor high electron-mobility transistors (MIS-HEMTs) have superior properties, including suppressed gate leakage current, large forward gate swing range [1,2], which is required by power switches in high-efficiency, high-speed power systems [3,4,5]. Different insulators (e.g., Al_2_O_3_, HfO_2_, SiO_2_, AlN and SiN_x_) [6,7,8] have been used as AlGaN/GaN MIS-HEMTs gate dielectric. The atomic layer-deposited (ALD) Al_2_O_3_ is more preferred because of its larger conduction band offset, high dielectric constant and high breakdown field values [9,10,11]. However, it has been reported that there is a large amount of hydroxyl (-OH) groups in ALD-Al_2_O_3_ [12,13] that use trimethylaluminum (TMA) and water as precursors. These -OH groups act as trap states and cause the AlGaN/GaN MIS-HEMTs to suffer from serious gate reliability and threshold voltage (*V*_th_) instability challenge [14,15].

It is suggested that using O_3_ as an oxidant during the deposition process of Al_2_O_3_ can improve device performance [16], but the carbon impurity in Al_2_O_3_ film increases [17]. It has been reported that there is less trap state density in the O_2_ plasma-assisted ALD-Al_2_O_3_ film [18,19]. It has been reported that adding O_2_ plasma in each ALD cycle can improve Al_2_O_3_ film quality [20]. However, the AlGaN surface can be damaged by O_2_ plasma at the initial stage of Al_2_O_3_ film deposition, since the O_2_ plasma can introduce deep-level traps at the AlGaN surface, which leads to device performance degradation and current collapse [21]. Meanwhile, there is little research on the threshold stability and gate reliability of the ALD-Al_2_O_3_ gate dielectric. We have already characterized the trap states and performance of the device with alternating O_2_ plasma treatment in our previous articles [22]. In this work, the *V*_th_ stability and gate reliability characteristics of the AlGaN/GaN MIS-HEMTs with the alternating O_2_ plasma treatment were investigated.

## 2. Device Structure and Fabrication Process

The AlGaN/GaN MIS-HEMTs were fabricated on the AlGaN/GaN heterostructure epitaxial sample, which was grown by metal–organic chemical vapor deposition (MOCVD). It consists of a 20 nm undoped Al_0.23_Ga_0.77_N barrier layer, 180 nm unintentionally doped GaN channel layer and 5.1 μm C-doped GaN buffer layer grown using MOCVD on a 6-inch Si (111) substrate. Figure 1a shows the schematic cross-sectional illustration of the AlGaN/GaN MIS-HEMTs. The AlGaN/GaN MIS-HEMTs process began with AlN/SiN_x_ passivation layer deposition. The device active region was isolated by mesa etching using BCl_3_/Ar. Then, a Ti/Al/Ni/Au metal stack with a thickness of 20/160/50/50 nm was deposited by Electron Beam Evaporation (EBE) on the source/drain region, and ohmic contact was achieved by rapid thermal process (RTP) at 780 °C for 30 s in N_2_ ambient. The transfer length method (TLM) test results show that the contact resistance was 1Ω·mm. The ALD-Al_2_O_3_ gate dielectrics with and without the alternating O_2_ plasma treatment were deposited and denoted as devices A and B, respectively. Finally, Ni/Au metal stack was deposited for the gate electrode.

The schematic process flow of depositing ALD-Al_2_O_3_ film with the alternating O_2_ plasma treatment is shown in Figure 1b. The entire depositing process was carried out using a Sentech SI ALD system. The deposition process consisted of a cycle of two sub-processes. Sub-process one: 4 nm ALD-Al_2_O_3_ was deposited with TMA and H_2_O as precursors. Sub-process two: The film deposited in sub-process one was treated with in situ O_2_ plasma for 2 min. The O_2_ gas flow was 100 sccm, gas pressure was 15 Pa, and the plasma power was 100 W. Throughout the process, the substrate temperature was maintained at 300 °C. Sub-process one and sub-process two were repeated five times. Finally, a 20 nm ALD-Al_2_O_3_ film with alternating O_2_ plasma treatment was obtained. It is worth noting that the deposited 4 nm ALD-Al_2_O_3_ film could serve as a protective layer on the AlGaN surface to prevent the O_2_ plasma damage [23].

## 3. Results and Discussion

Figure 2 exhibits the atomic force microscopy (AFM) image of the Al_2_O_3_ film surface with an area of 2 µm × 2 µm. For the Al_2_O_3_ film with and without the alternating O_2_ plasma treatment, the root mean square (RMS) of surface roughness is 0.094 nm and 0.096 nm, respectively. This indicates that the alternating O_2_ plasma treatment will not have adverse effects on the surface morphology of the Al_2_O_3_ film.

The gate leakage current density of the device is illustrated in Figure 3a. The gate leakage density of device A significantly decreased compared with that of device B. The breakdown voltage of device A also improved. In order to explore the reasons for the reduction of gate leakage in device A, the gate leakage mechanism was analyzed. Considering that ALD-Al_2_O_3_ has good quality, Fowler–Nordheim (FN) tunneling was believed to be the dominant gate leakage mechanism [24]. The effective barrier width of the dielectric narrowed under the forward gate voltage, and driven by the electric field in the gate dielectric, electrons at the Al_2_O_3_/AlGaN interface could directly tunnel through the gate dielectric. Leakage current by FN tunneling is illustrated in Figure 3b, which can be expressed as
(1)$$J_{FN}=\frac{q^{2}}{16\pi {\hbar \varphi }_{ox}}E_{ox}^{2}exp(-\frac{4\sqrt{2m*{\left(q\varphi_{ox}\right)}^{3}}}{3\hbar qE_{ox}})$$
where *q* is the charge of electrons, ℏ is the Planck’s constant, $\varphi_{ox}$ is the conduction band offset at Al_2_O_3_/AlGaN interface, $E_{ox}$ is the electric field strength in Al_2_O_3_ gate dielectric, $m^{*}$ is the effective electron mass in Al_2_O_3_, and 0.23${\;m}_{0}$ of an electron mass was used for the Al_2_O_3_ film [19]. The FN plots of log $\left(J/E_{ox}^{2}\right)$ versus $1/E_{ox}$ were straight lines, as shown in Figure 3b, indicating that FN tunneling was the dominant gate leakage mechanism under a high electric field. The linear slope was used to extract the conduction band offset at the Al_2_O_3_/AlGaN interface, which were 2.40 and 1.87 eV, respectively, for devices A and B. The lower gate leakage current density of device A was attributed to the higher conduction band offset at the Al_2_O_3_/AlGaN interface. The conduction band offset for device A was larger than the value of 2.2 eV in Ref. [25].

Time-dependent dielectric breakdown (TDDB) is one of the most common characterization methods for evaluating gate dielectric reliability [26]. The testing process of TDDB involves applying a constant bias stress on the gate dielectric for a long time, and monitoring the variation in leakage current passing through the dielectric layer. The quality of the gate dielectric can be evaluated using the magnitude of leakage and the time to breakdown (*t*_BD_) under the same gate bias stress. The reasons for leakage current and breakdown of the gate dielectric are as follows. There are defects inside the gate dielectric at the initial state, and these defects are mainly bulk defects formed during the sedimentation process. Applying electrical stress to the gate dielectric can induce random defects within the gate dielectric, causing leakage current. In addition, when electrons accelerate through the gate dielectric, it can also cause damage to the gate dielectric and form new defects. When the defects form a continuous seepage path inside the gate dielectric, the leakage current rapidly increases and the gate dielectric layer undergoes breakdown. High electrical stress will accelerate the generation of defects, generate higher leakage current, and thus accelerate the breakdown process of the gate dielectric. Due to the different breakdown voltages for device A and device B, two sets of gate bias were used to stress the devices A and B, respectively. The time-dependent gate breakdown characteristics are shown in Figure 4a,b. The *t*_BD_ for gate dielectric at different gate voltages statistically obey the Weibull distribution, which can be described by [27]
(2)$$F(t)=1-exp[-{\left(\frac{t}{\eta }\right)}^{\beta }]$$
where *t* is the gate voltage application time, β is the Weibull slope, and η is the characteristic lifespan or scale factor.

The Weibull failure distribution is linearly simplified as follows:(3)ln[−ln(1−F(t))]=βln(t)−βln(η)

A larger β indicates a more concentrated distribution of *t*_BD_ in the breakdown characteristic [28]. Figure 4c,d shows the Weibull plots of the *t*_BD_ distribution for devices A and B. Weibull slope β was extracted and found to be 5 and 4.5 for devices A and B, which indicated that ALD-Al_2_O_3_ with alternating O_2_ plasma treatment has better quality and reliability. These results were larger than the value of 4.45 in Ref. [29], although Al_2_O_3_ had a thicker thickness (25 nm).

The *V*_th_ instability induced by high-temperature operation and long-term gate stress limits the commercial application of AlGaN/GaN MIS-HEMTs. To investigate the thermal stability of *V*_th_, the transfer characteristic curves of device A and device B at various temperatures from 30 °C to 150 °C in steps of 30 °C were measured, as is shown in Figure 5. The OFF-state drain current increased by about two orders of magnitude as a result of increased buffer leakage current [30]. The ON-state *I*_DS_ decreased slightly due to the lower carrier mobility at higher temperatures [31].

Figure 6 shows the temperature-dependence *V*_th_ shift (Δ*V*_th_) for devices A and B. The device A demonstrated a better *V*_th_ thermal stability and the maximum Δ*V*_th_ of 0.24 V was achieved at 150 °C at the *I*_DS_ level of 1 μA/mm, less than that of 0.55 V for device B at 150 °C. However, Δ*V*_th_ in Ref. [30] is larger than 1V at the same test temperature.

To assess the *V*_th_ stability of the device under long-time gate bias stress, the forward gate bias stress (*V_G_*__stress_) of 2 V was applied to the gate with source and drain grounded. A quick *I*_D_-*V*_GS_ test was conducted after certain interval times (1, 5, 10, 20, 40, 60, 80, 100, 200, 400, 500, 600, 800, 1000, 2000, and 3000 s). Figure 7 shows the multiple *I*_D_-*V*_GS_ curves throughout the entire testing process. The *I*_D_-*V*_GS_ curves positively shift under the forward gate bias stress, which corresponded to electrons in the channel being trapped [32]. During the forward gate bias stress application process, the electric field in the AlGaN barrier layer is very high, especially at the edge of the gate. A strong electric field can cause electrons to tunnel from the defects in the AlGaN barrier layer to the valence band, which is known as Zener trapping. Electrons in 2DEG are then emitted into the defects, causing a decrease in electron concentration in the channel and a positive shift in the *V*_th_.

As shown in Figure 8, the extracted Δ*V*_th_ after the 3000 s gate bias stress of 2 V were 0.55 V and 0.88 V for devices A and B, respectively. Device A showed a relatively small Δ*V*_th_ compared to device B. This indicated that the trap state density in the dielectric was reduced by the alternating O_2_ plasma treatment [15]. Furthermore, subthreshold slope (SS) did not show any significant changes after long-time gate bias stress for both devices.

## 4. Conclusions

The *V*_th_ stability and gate reliability of the AlGaN/GaN MIS-HEMTs with alternating O_2_ plasma treatment were investigated in this article. The conduction band offset at the Al_2_O_3_/AlGaN interface was elevated to 2.4 eV after the alternating O_2_ plasma treatment, and hence resulted in lower gate leakage current density. The gate dielectric reliability was also improved, which was characterized by the TDDB test. The device with the alternating O_2_ plasma treatment also showed improved thermal stability of *V*_th_ and long-time gate bias induced *V*_th_ instability. The proposed O_2_ plasma alternating treatment technique was found to exhibit superior performance, which is highly desirable in high-performance and reliable power devices.

## Figures and Tables

**Figure 1 nanomaterials-14-00523-f001:**
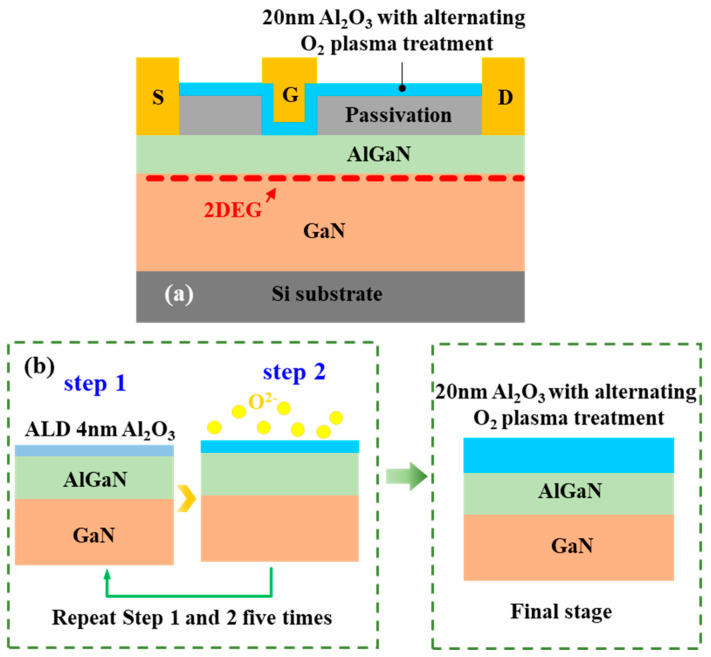
(**a**) Schematic cross-sectional illustration of the AlGaN/GaN MIS-HEMT. (**b**) Schematic process flow of depositing ALD-Al_2_O_3_ film with the alternating O_2_ plasma treatment.

**Figure 2 nanomaterials-14-00523-f002:**
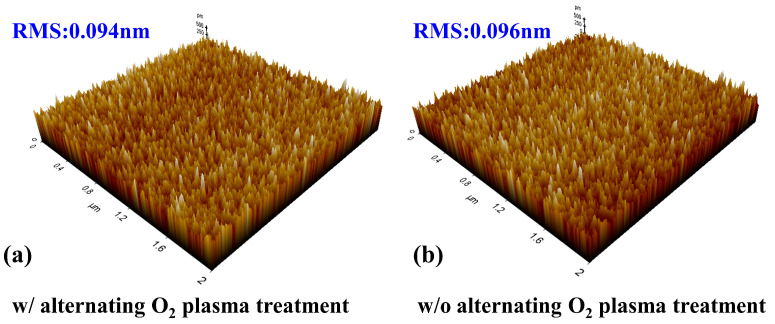
(**a**,**b**) 2 µm × 2 µm surface morphology of the ALD-Al_2_O_3_.

**Figure 3 nanomaterials-14-00523-f003:**
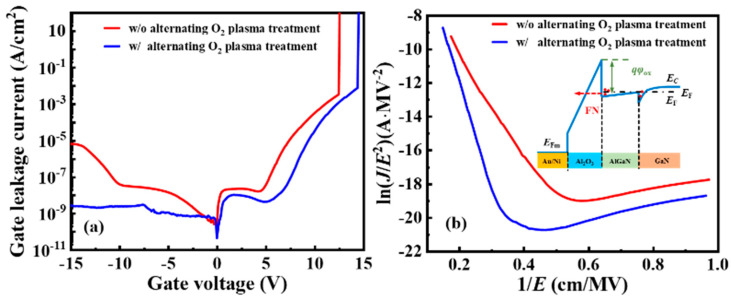
(**a**) Gate leakage current density characteristic and (**b**) FN tunneling plot of log $\left(J/E_{ox}^{2}\right)$ versus $1/E_{ox}$ for device A and device B.

**Figure 4 nanomaterials-14-00523-f004:**
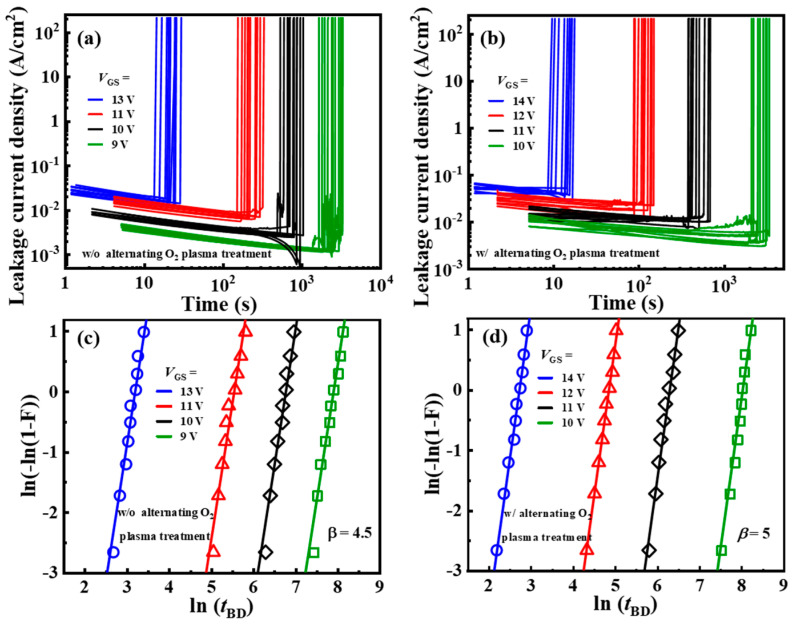
(**a**,**b**) *t*_BD_ of device A and device B. (**c**,**d**) Weibull plots of the *t*_BD_ distribution for device A and device B.

**Figure 5 nanomaterials-14-00523-f005:**
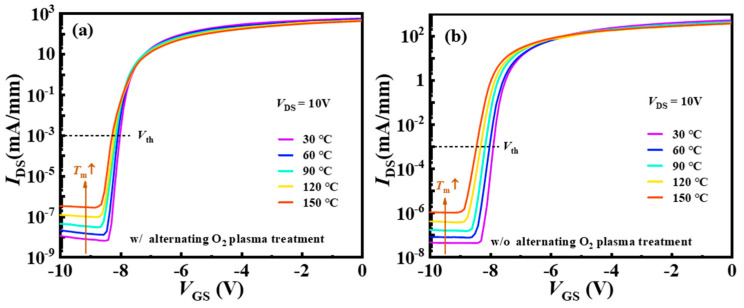
(**a**,**b**) Temperature-dependent *I*_D_-*V*_GS_ characteristics of device A and device B with the measurement temperature increasing from 30 to 150 °C.

**Figure 6 nanomaterials-14-00523-f006:**
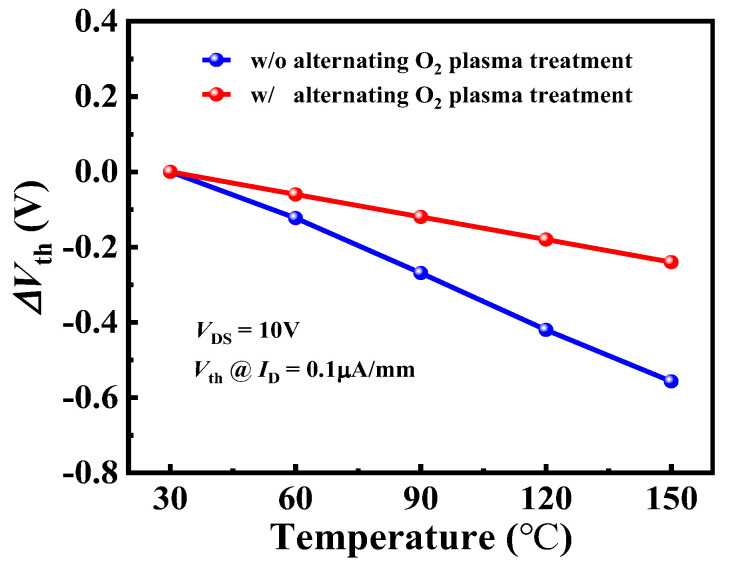
The measured temperature-dependent Δ*V*_th_ for device A and device B.

**Figure 7 nanomaterials-14-00523-f007:**
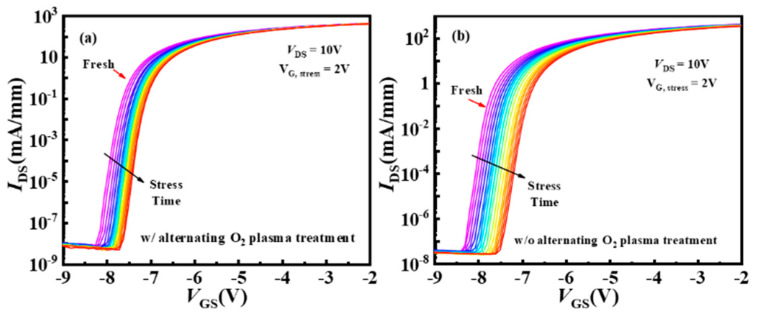
(**a**,**b**) Multiple *I*_D_-*V*_GS_ characteristics of the MIS-HEMTs during the 3000 s gate bias stress of 2 V for device A and device B.

**Figure 8 nanomaterials-14-00523-f008:**
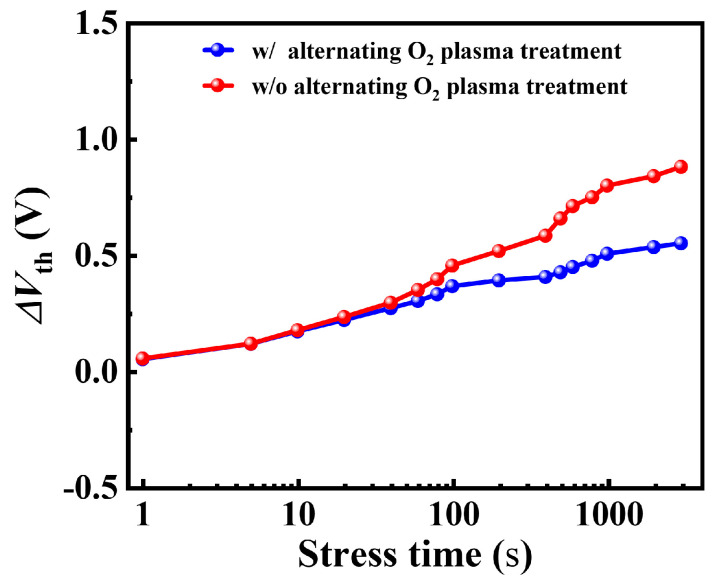
The measured Δ*V*_th_ during the 3000 s forward gate bias stress for device A and device B.

## Data Availability

Data are contained within the article.

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
