# Peer review of "Improved Vth Stability and Gate Reliability of GaN-Based MIS-HEMTs by Employing Alternating O2 Plasma Treatment"

_nanomaterials, 2024, doi:10.3390/nano14060523_

Round 1

Reviewer 1 Report

Comments and Suggestions for Authors

Although the topic of the manuscript is very interesting, I have some comments:

1) My first comment is on the match between the manuscript and the aim and scope of Nanomaterials. In the work there are not nanomaterials in terms of nanocrystals, nanolithographic designs etc. I see that some layer in the device and at a nanometre length scale. It would be nice if the authors shed light on this point. Is the nanometric scale of some components of the device playing a significant role?

2) In the title and in the abstract it should be specified that MIS-HEMTs are "metal-insulator-semiconductor high electron-mobility transistors".

3) No morphological characterization (electron or probe microscopy, for example) is presented in the work and this would be highly beneficial.

4) X-ray photoelectron spectroscopy and EDX would be very beneficial to characterize the deposition processes.

5) Can the authors say something on the ambipolarity of the active material in the transistor?

6) The authors better specify the choice of the GaN/AlGaN junction for this study?

Reviewer 2 Report

Comments and Suggestions for Authors

This article discusses the threshold voltage shift in GaN MIS-HEMTs with and without alternating O2 plasma treatment. The article is quite focused and well-posed.

The authors are surely aware that traps also impact the dynamic behavior of GaN devices. Dynamic on-resistance changes and threshold shifts are well-known phenomena, so it would be interesting to know how this treatment could impact a dynamic operating regime. Tests such as those conducted by Alemanno, A, et al. in "A Reconfigurable Setup for the On-Wafer Characterization of the Dynamic RON of 600 V GaN Switches at Variable Operating Regimes" (Electronics, 2023) could be performed in this regard. I understand that the authors might not have immediate capabilities to implement these tests, but could the author introduce a comment in this regard? What would be expected?

At line 42, the authors mention that they have already characterized the trap states and DC characteristics of devices with and without alternating O2 plasma treatment in previous articles. Please specify and reference the precise articles. In general, I would advise the authors to better position their contribution within the state-of-the-art.

Round 2

Reviewer 1 Report

Comments and Suggestions for Authors

The revisions are proper.